# Monitoring and Modulating Diet and Gut Microbes to Enhance Response and Reduce Toxicity to Cancer Treatment

**DOI:** 10.3390/cancers15030777

**Published:** 2023-01-27

**Authors:** Anne Knisely, Yongwoo David Seo, Jennifer A. Wargo, Manoj Chelvanambi

**Affiliations:** 1Department of Gynecologic Oncology and Reproductive Medicine, The University of Texas MD Anderson Cancer Center, Houston, TX 77030, USA; 2Department of Surgical Oncology, The University of Texas MD Anderson Cancer Center, Houston, TX 77030, USA

**Keywords:** gut microbiome, cancer immunotherapy, microbiome modulation, antibiotics, toxicity, dietary interventions

## Abstract

**Simple Summary:**

The gut microbiome has been shown to play a role in carcinogenesis and the progression of cancer, in part through its interaction with the host immune system. Research from numerous clinical cohorts and preclinical models suggests that gut microbes contribute to response and toxicity to cancer treatment—including chemotherapy, immunotherapy, and radiation. Furthermore, disrupting the gut microbiome with broad spectrum antibiotics negatively impacts the outcomes to cancer therapy. Studies have shown improved oncologic outcomes to immunotherapy and other treatment in the setting of specific dietary patterns, such as a high fiber diet. Accordingly, therapeutic strategies including fecal microbiome transplant, pre/probiotics, and dietary interventions have emerged aiming to improve patient outcomes and are being tested in ongoing clinical trials. The aim of the present work is to provide an update on the available evidence regarding how gut microbes and other factors affect the response and toxicity to cancer therapy, with opportunities to target these therapeutically.

**Abstract:**

The gut microbiome comprises a diverse array of microbial species that have been shown to dynamically modulate host immunity both locally and systemically, as well as contribute to tumorigenesis. In this review, we discuss the scientific evidence on the role that gut microbes and diet play in response and toxicity to cancer treatment. We highlight studies across multiple cancer cohorts that have shown an association between particular gut microbiome signatures and an improved response to immune checkpoint blockade, chemotherapy, and adoptive cell therapies, as well as the role of particular microbes in driving treatment-related toxicity and how the microbiome can be modulated through strategies, such as fecal transplant. We also summarize the current literature that implicate high fiber and ketogenic diets in improved response rates to immunotherapy and chemotherapy, respectively. Finally, we discuss the relevance of these findings in the context of patient care, advocate for a holistic approach to cancer treatment, and comment on the next frontier of targeted gut and tumor microbiome modulation through novel therapeutics, dietary intervention, and precision-medicine approaches.

## 1. Brief Introduction

The gut microbiome comprises a diverse and complex array of microbes that play integral roles in regulating the immune system and maintaining human health. Alterations in the composition and diversity of the gut microbiota are associated with a variety of molecular and physiologic changes, and such alterations have been implicated in multiple disease processes, including cancer—specifically oncogenesis, tumor progression, and response to therapy [1]. There is an emerging body of evidence that highlights the importance of the gut microbiome composition in the efficacy, as well as toxicity, of various cancer treatments, including immunotherapy [2,3,4,5,6,7,8,9], chemotherapy [10,11,12], and targeted and adoptive cell therapies [13,14]. Given these findings, there is significant interest in modulating the gut microbiome to affect treatment responses. Several interventions, including fecal microbiome transplant, prebiotic, probiotic, and antibiotic therapies, as well as dietary intervention strategies, have emerged and shown promising results as strategies for modulating the microbiome.

In this review, we will summarize the key findings to highlight the emerging relationship between the gut microbiome and response, resistance, and toxicity observed with various cancer treatments. Further, we also outline the current and possible future strategies that seek to modulate the gut microbiome to prospectively improve clinical outcomes to existing therapies in future trials.

## 2. The Role of the Gut Microbiome in Response to Cancer Treatment

### 2.1. Immunotherapy

Gut microbes have been shown to modulate host immunity in multiple important ways—locally at the level of the gut mucosa, via crosstalk between gut commensals and mucosal immune cells, and systemically via immune cell priming, among other mechanisms [15]. These gut resident microbes maintain a dynamic relationship with the host immune system [16,17], and different microbiome compositions have been shown to effect the host immune tone [18] and ultimately may affect the response to immunotherapy in cancer.

Immune checkpoint inhibitors (ICIs) have proven to be highly effective against multiple solid tumors, including advanced melanoma, non-small cell lung cancer (NSCLC), and renal cell carcinoma (RCC) [19,20,21,22,23]. However, resistance is observed in a substantial portion of patients who do not achieve durable responses [24]. Thus, novel biomarkers of response and resistance are critical for understanding the underlying mechanisms and in devising novel therapies aimed at improving outcomes in future trials. Evidence from several clinical cancer cohorts suggests that unique gut microbial taxa/signatures may indeed characterize response and/or resistance to ICI.

Hallmark studies in melanoma first established such an association between the composition of the gut microbiome and the response to programmed cell death protein 1 (PD-1) blockade [3,5]. In one study, 112 patients with metastatic melanoma treated with anti-PD-1 therapy observed that the gut microbiota of responders (*n* = 30) harbored significantly greater compositional diversity when compared to non-responders (*n* = 13), with specific enrichment of taxa from the family *Ruminococcaceae* [3]. These trends were also observed in an independent cohort of hepatocellular carcinoma patients [3,8]. *Ruminococcaceae* and *Bacteroidaceae*-dominated microbiomes were also associated with an improved response to ICI in metastatic melanoma patients in Australia and the Netherlands [25]. Separately, in another metastatic melanoma patient cohort (*n* = 42, overall response rate (ORR) = 38%) [5], and two NSCLC cohorts [7,9], members of the Bifidobacteriaceae family, including *Bifidobacterium longum*, were shown to be enriched in ICI-responders vs. non-responders.

Furthermore, in a study examining the ICI response in patients with NSCLC (*n* = 60) and RCC (*n* = 40), researchers found that primary resistance to ICIs can be attributed to an abnormal gut microbiome composition [2]; the commensal that was most significantly associated with a favorable clinical outcome in their study (in both cancer groups) was *Akkermansia muciniphila*, which was enriched in patients with the best clinical response and longest progression-free survival prior to the initiation of treatment. *A. muciniphila* has also been shown to be enriched in ICI responders in other cancer types, including hepatocellular carcinoma, highlighting its particular importance in response to ICI across multiple cancer patient populations [8].

However, the role of *A. muciniphila* as a definitive response promoting the gut microbe is confounded by recent data that suggests that overdominance of the gut microbiome by *A. muciniphila* could predispose poorer outcomes with ICI [6]. In a large cohort of advanced NSCLC patients treated with PD-1 blockade (*n* = 338), although a relative abundance of *Akkermansia* (Akk^+^) was associated with a 10% improvement in ORR, an overabundance of the *Akkermansia* (>77th percentile of Akk^+^ subgroup) was associated with shorter overall survival. This highlights the potential utility of precise quantification of Akk relative abundance to be used as a biomarker in this patient cohort for response to ICI.

A primary mechanism by which the gut microbiota modulate antitumor immunity is via metabolite production. For example, inosine, the purine metabolite produced by *A. muciniphilia* and *Bifidobacterium pseudolongum*, among other species, has been found to enter systemic circulation and promote Th1 activation, thereby improving the efficacy of ICI [26]. Other gut microbe metabolites, including short-chain fatty acids and anacardic acid, have also been shown to modulate antitumor immune responses [27].

The current research demonstrates that although microbes serve as important biomarkers, heterogeneity in response-associated signatures exists. This suggests that different families of resident gut bacteria may converge at the level of the physiologic function and ultimately exert their influence to promote ICI response through the secretion of metabolites that serve key functions in immune system modulation.

### 2.2. Chemotherapy

Host microbes also influence the response to chemotherapy in a context-specific manner, as different taxa have been shown to exert different effects on outcomes to chemotherapy.

On one hand, the presence of particular bacterial species in the intratumoral microbiome can undermine the effect of chemotherapeutic drugs by way of modification of the active agent by prokaryotic enzymes [11,28,29]. For example, in an in vitro colon cancer model, the presence of an intact intratumoral microbiome was sufficient to modify gemcitabine, an antimetabolite chemotherapy agent used to treat various cancer types, into its inactive form to lower its cytotoxic potency. Subsequent mechanistic studies demonstrated that this effect was remarkably dependent on the expression of a long isoform of the bacterial enzyme cytidine deaminase (CDD_L_), which was expressed primarily by members of Gammaproteobacteria. This axis of resistance was then validated in preclinical colon cancer mouse models where the use of the antibiotic, Ciprofloxacin, reversed resistance to gemcitabine. Elegant findings from these studies thus confirmed and highlighted the important roles played by the host microbiome in altering the response to chemotherapy [11].

Contrastingly, however, other preclinical studies have demonstrated that the gut microbiome may indirectly enhance the response to chemotherapy. Cyclophosphamide, an alkylating anticancer agent, affects the composition of the small intestine microbiome via reduction of the bacterial species of the *Firmicutes* phylum and promotes disruption of the intestinal barrier with translocation of certain bacteria, including, but not limited to segmented filamentous bacteria and *Lactobacillus johnsonii*, *Lactobacillus murinus*, *Barnesiella intestinihominis*, and *Enterococcus hirae*, into secondary lymphoid organs, where these microbes strengthen immune priming and specific anti-tumor immunity by stimulating helper T-cell function. Interestingly, treatment with broad-spectrum antibiotics (vancomycin and colistin) resulted in developed resistance to cyclophosphamide with reduced frequencies of tumor-infiltrating CD3+ T-cells and Th1 cells, thus highlighting the overall importance of the gut microbiota in enhancing the response to chemotherapy [10,30].

In addition to directly affecting the outcomes to chemotherapy, the composition of the gut microbiome may also serve as a prognostic indicator in the setting of treatment with chemotherapeutic agents. A recent study of patients with epithelial ovarian cancer demonstrated that distinct signatures of gut microbiota characterize an exceptional response and observed resistance to platinum-based chemotherapy. While the gut microbiome of non-responders showed reduced overall diversity and specific enrichment of taxa such as *Coriobacteriaceae* and *Bifidobacterium*, an exceptional response to platinum-based chemotherapy was associated with increased compositional diversity and enrichment of lactate-utilizing microbes belonging to the *Veillonellaceae* family. These findings highlight the predictive/prognostic role of intestinal microbiota in evaluating and monitoring the clinical response to ovarian cancer therapy [12].

### 2.3. Targeted and Other Therapies

With the growing incorporation of targeted therapies and adoptive cell therapy techniques into cancer treatment, researchers have begun to investigate the role the gut microbiome may play in modulating responses to these alternative treatment modalities.

Chimeric antigen receptor (CAR) T-cell therapy directed against CD19 has shown significant success in the treatment of B-cell leukemia and lymphoma [31,32,33]. A recent study published by Smith et al. analyzed whether particular gut microbiome compositions were associated with improved clinical outcomes after CD19 CAR T-cell therapy in patients with B-cell malignancies; they found that selected bacterial taxa *Ruminococcus*, *Bacteroides*, and *Faecalibacterium* were associated with day 100 complete response, whereas *Veillonellaceae* was found in higher abundance in patients with lower day 100 complete response rates [13]. Another study focusing on patients with hematologic malignancies sampled the fecal microbiome of over one thousand patients undergoing allogeneic hematopoietic-cell transplantation at four centers and demonstrated that higher diversity of intestinal microbiota at the time of neutrophil engraftment was associated with lower mortality [34].

It is increasingly becoming more common for early phase clinical trials to include translational endpoints, including microbiome investigations. A phase Ib/II study of regorafenib, a multi-kinase inhibitor of vascular endothelial growth factor (VEGF) receptors, and toripalimab, an anti-PD-1 monoclonal antibody (mAb), in patients with metastatic colorectal cancer demonstrated a relatively low objective response rate of 15.2%. Correlative microbiome studies show a significantly increased abundance of *Fusobactrium* in non-responders compared to responders, representing a potential target for treatment modulation in this population [14]. A full list of clinical trials examining the gut microbiome and response to cancer treatment is available in Table 1.

### 2.4. Impact of Antibiotics

Antibiotic use among cancer patients has increased over time and while this may contribute to decreased mortality from an infection standpoint, antibiotic use can impart rapid and long-lasting effects on gut microbiota composition, leading to decreased alpha diversity, metabolic capacity changes, loss of vital taxa, and impaired cytotoxic T-cell response against cancer [38,39].

Preclinical studies have investigated the role of antibiotic treatment on cancer treatment response via intermediary effects on the gut microbiome. In lymphoma, colon cancer, and melanoma mouse models, the receipt of an antibiotic cocktail (vancomycin, imipenem, and neomycin) resulted in a poor response to immunotherapy and platinum chemotherapy secondary to poorly functioning myeloid-derived cells in the tumor microenvironment [40].

A number of clinical studies have also demonstrated negative effects of antibiotic use on oncologic outcomes, with the most research surrounding the efficacy of ICI. One study found that patients with advanced NSCLC or urothelial carcinoma who were prescribed antibiotics (beta-lactam inhibitors, fluoroquinolones, or macrolides) within 2 months prior to the initiation of treatment with PD-1/PD-L1 mAb, demonstrated significantly shorter progression-free survival (PFS) and overall survival (OS) [2]. Another study in patients (*n* = 196) with multiple cancer types (NSCLC, melanoma, and others) demonstrated worse overall survival in patients who received broad-spectrum antibiotics prior to ICI therapy, independent of tumor site, disease burden, and performance status [39]. Similar findings have also been replicated in multiple tumor types for patients treated with ICI [41,42]. Specifically in urothelial carcinoma, in a post-hoc analysis of multiple clinical trials, antibiotic use within 30 days of treatment initiation was associated with worse OS and PFS with atezolizumab, but not chemotherapy, suggesting that the negative effect of antibiotics may be specific to immunotherapies [43]. Similarly, analysis of a phase II clinical trial of advanced renal cell carcinoma patients treated with nivolumab showed a reduction in the objective response rate from 28% to 9% in patients who had recent antibiotic use, with a significant effect on the microbiome composition [44].

Further studies are needed to examine the impact of antibiotics on cancer treatment efficacy, especially in other non-immunotherapy treatment modalities, as this represents a simple and feasible intervention that could vastly improve patient outcomes.

### 2.5. Modulating Gut Microbes to Improve Outcomes—The Use of Fecal Microbiome Transplantation

Modulation of the gut microbiome can be accomplished in a variety of ways, including the introduction of antibiotics (as discussed above), probiotics and prebiotics, dietary interventions, and fecal microbiome transplantation (FMT).

A number of preclinical studies in metastatic melanoma have demonstrated improved oncologic outcomes following FMT [3,5]. Seminal work by Gopalakrishnan et al. demonstrates that FMT in preclinical models using stool from ICI responders or non-responders as donors imparts different effects on the local anti-tumor immunity and systemic inflammation. Specifically, transplantation of a ‘favorable’ microbiome from ICI responders was associated with a significant enrichment of the innate effector cells and CD8+ T-cells and a concomitant decrease in the intratumoral frequency of suppressive myeloid cells when compared to the tumor microenvironment (TME) of tumor-bearing mice receiving FMT from ICI non-responders [3].

Furthermore, results from a first-in-human clinical trial to assess the effect of FMT on the response to anti-PD-1 immunotherapy in a metastatic melanoma population were recently published [45]. In this trial, the researchers used two FMT donors who had achieved a complete response for at least one year after treatment with ICI monotherapy and observed that three out of 10 recipients (patients who had previously progressed on at least one line of anti-PD-1 therapy) demonstrated safe and objective clinical responses. These clinical responses with FMT were also associated with favorable changes in the immune cell infiltrates in the gut (increased lamina propria infiltration of CD68+ antigen presenting cells) and in the TME (increased posttreatment intratumoral CD8+ T cell infiltration) [45].

Together, these studies demonstrate that manipulation of the gut microbiome via FMT has clinical promise as a strategic intervention to improve oncologic outcomes, especially in patients receiving immunotherapy.

## 3. The Role of the Diet in Response to Cancer Treatment

Nutrition and diet can affect tumor growth via local effects within the tumor microenvironment, regional effects via modulation of the gut microbiome, and systemic immune effects. Obesity, which has been found to be associated with many cancer types, represents a systemic inflammatory condition characterized by increased production of interleukin (IL)-17 and IL-21, which are strong inducers of Th17 cells, leading to a potential imbalance of Treg and Th17 cells [46]. A number of observational studies have demonstrated both decreased cancer prevalence and mortality associated with adherence to particular fiber-rich diets, such as Mediterranean diets [47,48,49,50,51], which have been linked to lower systemic inflammation and overall enhanced immune function of cytotoxic and T-helper cells [52].

### 3.1. Immunotherapy

There is growing interest in studying the role of the diet in response to cancer immunotherapy, given the proven impact of microbiome differences in this setting and the knowledge that diet is one strategy to modulate the microbiome.

Recently, a landmark study by Spencer et al. demonstrated that increased dietary fiber intake and probiotic use may have a significant impact on clinical outcomes in patients with metastatic melanoma treated with ICI [53]. By concurrently studying fiber intake and pathologic response to ICI in 128 patients with metastatic melanoma, the authors found that patients who reported sufficient dietary fiber intake (defined as >20 g per day) demonstrated improved odds of response to ICI and improved PFS over those with insufficient dietary fiber intake (<20 g per day) (median PFS not reached versus 13 months), with every 5 g increase in fiber intake corresponding with a 30% lower risk of progression of disease or death. Further, by only modifying the fiber intake of mice in follow-up preclinical studies, the authors observed distinct gut microbiome profiles and significantly delayed tumor growth on-treatment with ICI in mice that received a fiber-rich diet compared to mice receiving a low-fiber diet. Importantly, these observations were microbiome-dependent, as these beneficial effects were not recapitulated in germ-free mice. Furthermore, deep immune profiling of the TME demonstrated that fiber intake was directly related to the strength of the intratumoral interferon gamma (IFNγ)+ T-cell response which, although warranting further investigation, highlights one associated mechanism through which such dietary interventions might promote a response to ICI [53]. In another melanoma cohort, the consumption of fiber and omega 3 fatty acids was associated with enhanced microbial diversity and enrichment of genus *Ruminococcaceae*, which have both been associated with an improved response to ICI therapy in other studies [25].

Another study that evaluated the effect of targeted dietary interventions on ICI response specifically looked at oral supplementation with the polyphenol-rich berry camu-camu (CC), also known as *Myrciaria dubia*, which has been shown to be protective against metabolic disorders in mice through the enrichment of *A. muciniphila* and *Bifidobacterium* in the gut [54,55]. In preclinical models, the authors observed delayed tumor growth in mice (in sarcoma and breast cancer models) that were treated with combined CC and α-PD-1 compared to mice that only received α-PD-1 treatment alone. In these studies, the polyphenol castalagin was identified as the active anti-tumor ingredient in the berry [54]. Further investigation into the mechanism of the action of this specific compound in ICI treated preclinical mouse studies and prospective human cancer trials is warranted.

Despite the limited number of published studies evaluating the role of the diet in response to cancer immunotherapy specifically, this is an important area of research as we work towards novel ways to modulate the microbiome composition to improve outcomes for patients.

### 3.2. Chemotherapy and Other Therapies

A number of preclinical studies have shown the synergistic effects of fasting and caloric restriction on anticancer therapy, including radiation and chemotherapies, with slowed tumor growth demonstrated in breast, colorectal, melanoma, and glioma cancer models [56]. Fasting before and after chemotherapy administration has been shown to be safe and feasible in cancer patients [57,58]. A recent randomized phase 2 trial [59] investigated the effect of a fasting mimicking diet (FMD) at the time of neoadjuvant chemotherapy for breast cancer, based on in vitro and in vivo data that fasting renders cancer cells more sensitive to cancer therapy [60,61]. The investigators randomized 131 patients with human epidermal growth factor receptor 2 (HER2)-negative stage II/III breast cancer to FMD or a regular diet and found that a radiologically complete or partial response occurred more often in patients using the FMD. They also found a decrease in chemotherapy-induced DNA damage in T-cells in the FMD group, with no significant difference in the toxicity between the two groups. This study, although limited by suboptimal compliance of 33.8% of patients in the FMD group at 4 chemotherapy cycles, provides the first randomized data regarding the potential synergistic effect of fasting or a fasting mimicking diet during chemotherapy on cancer outcomes.

Alternatively, the use of a ketogenic diet, which may be better tolerated in some patients compared to fasting, has a long history of safety as an epilepsy treatment and has demonstrated safety in case studies of patients with gliomas [62]. The ketogenic diet has been shown to sponsor, through multiple mechanisms, an unfavorable metabolic environment for cancer cell proliferation, selectively starving cancer cells [63]. Preclinical studies utilizing murine models have demonstrated reduced tumor growth with a ketogenic diet, with synergistic effects observed with the combination of a ketogenic diet and radiation and/or chemotherapy in both glioma and lung cancer models [64,65,66]. In this regard, ongoing trials such as the diet restriction and exercise-induced adaptations in metastatic breast cancer (DREAM) trial (NCT03795493) aim to evaluate the therapeutic effect, measured as a function of the change in tumor burden as a primary outcome, of a short-term, 50% calorie-restricted and ketogenic diet combined with aerobic exercise during chemotherapy treatment for patients with metastatic breast cancer [67].

Furthermore, multiple studies from investigators in Turkey have evaluated the use of metabolically supported chemotherapy (MSCT), which involves a combination of fasting and administration of pharmacological doses of insulin to induce hypoglycemia at the time of standard chemotherapy, along with adherence to a ketogenic diet in advanced gastric, pancreatic, and lung cancer [68,69,70]. These studies demonstrated remarkably high PFS and OS rates, but notably did not include control groups. Therefore, controlled, comparative clinical trials are warranted to further investigate these interesting findings.

### 3.3. Modulating Diet to Improve Outcomes

It is increasingly evident that diet plays a critical role in modulating the gut microbiome and has a significant and observable impact on the anti-tumor immune response to immunotherapy. Based on the retrospective and preclinical data implicating the beneficial role of a high-fiber diet, multiple prospective randomized controlled trials are underway to attempt to demonstrate the efficacy of a high-fiber diet on improving the response and outcomes to checkpoint inhibitors across multiple tumor types (NCT04645680, NCT04866810, NCT04866810). Other studies are targeting specific diets, such as the ketogenic diet, for its ability to alter the response to checkpoint blockade, for instance in advanced renal cell carcinoma (NCT05119010) (Table 2).

The modulation of the gut microbiome balance utilizing pre- and probiotics is also underway, with multiple trials attempting to show a benefit in the response to a checkpoint blockade in a wide range of cancer types (NCT05032014, NCT04699721, NCT03829111). While there has been a trial studying the efficacy of the probiotic VSL#3 (containing live strains of *Lactobacillus* and *Bifidobacterium*) in improving the response to cancer therapy, this was in the setting of treatment with the EGFR inhibitor dacomitinib in non-small cell lung cancer [75]; to date, there has been no prospective trial evaluating the effect of probiotics on immunotherapy response.

While the prospective data generated from these studies have not yet matured, the next decade of translational research should see the transition of the field from multi-omic microbiome analysis and hypothesis generation to targetable interventions in dietary management for patients undergoing immunotherapy.

## 4. The Role of Gut Microbes and Diet in Toxicity

In addition to its impact on the response to cancer treatment, the gut microbiome and diet have also been implicated in the modulation of toxicity to therapy. Given its known ability to modulate the response to immune checkpoint inhibitors (ICI), the impact of the gut microbiome on immune-related adverse events (irAE) in response to ICI have also been explored in recent studies. irAEs are comprised of multiple different pathologies spanning every organ system, and while they can vary in severity, they are quite common, with a low grade irAE observed in over 90% of patients undergoing immune-modulating therapy for cancer [76,77,78]. The ability to effectively mitigate the potentially fatal course of irAEs have been hindered by an unpredictable time course and severity, the variability of irAE seen with different ICI combinations, and a lack of effective treatment besides cessation of therapy and initiation of high dose corticosteroids [76,79,80].

While the exact mechanistic underpinnings of irAE mediation remain incompletely understood, there has been increasing interest in evaluating the gut and host microbiome for both predicting irAEs and modulating toxicity to therapy. Seminal work in the setting of combination ICI treatment in advanced melanoma patients demonstrated that there was an upregulation of specific taxa such as *Bacteroides* within those who experienced significant toxicity to therapy; furthermore, there was evidence of upregulation of IL-1β in patient samples with immune-mediated colitis, as well as within preclinical models, suggestive of a mechanism by which unfavorable gut microbiota enhance the constitutive cytokine activation, which leads to irAEs [81].

There is also emerging evidence of gut microbe-mediated toxicity within treatment with immune agonist antibodies (IAAs), which are often limited in clinical use due to irAEs [82]. Preclinical models utilizing CD40 IAAs demonstrated that the presence of diverse gut flora leads to a MyD88-dependent activation of the host immune system, especially macrophages, which results in the rapid production of inflammatory cytokines, such as tumor necrosis factor alpha (TNFα), IL-6, and IFN-I and an acute induction of macrophage- and neutrophil-dependent liver damage [83]. Interestingly, toxicity observed with CD137 IAAs also appears to converge at the level of host MyD88 activation by gut microbiota, which results in a CD8+ T-cell-dependent liver damage and IFNγ driven systemic inflammation. Importantly, toxicity to IAA therapy was reduced in germ-free or antibiotic-treated mice in this model without a deleterious impact on antitumor immunity, suggesting a potential roadmap for irAE modulation and prevention in the clinical setting [83,84].

Similarly, chimeric antigen receptor (CAR) T-cell therapy, while revolutionizing the treatment of certain hematologic malignancies, has also been hindered by systemic inflammation and specific irAEs (such as immune effector cell-associated neurotoxicity syndrome, or ICANS) with few available biomarkers to predict the severity [85]. Recent work utilized a retrospective review of patients with non-Hodgkin lymphoma (NHL) or acute lymphocytic leukemia (ALL) treated with CD19 CAR T-cells and found that the administration of broad spectrum antibiotics within four weeks prior to the first treatment was strongly associated with shorter overall survival and increased incidence of ICANS [13]. A prospective smaller cohort of NHL or ALL patients with matched baseline microbiome profiling found that enrichment of specific bacterial taxa such as *Bacteroides*, *Ruminococcus* and *Faecalibacterium* was associated with not only response to therapy, but a toxicity-free status, further suggesting the importance of gut dysbiosis as a predictor of toxicity to immune-mediated therapies [13,86].

There is some evidence of the importance of a diverse gut microbiome within the context of toxicity to traditional therapies such as chemoradiation; higher gut microbial diversity has been associated with decreased toxicity to chemoradiation for cervical cancer, as well as in the setting of pelvic radiotherapy alone [87,88]. Fecal microbiota transplantation (FMT) from healthy donors has been utilized with some success in the clinical setting to improve toxicity to both cytotoxic therapy and to radiotherapy [89,90].

The gut microbiome has also been implicated in the modulation of graft-versus-host disease (GVHD) in allogeneic stem cell transplantation (SCT) performed for the treatment of hematologic malignancies. GVHD is mediated by the donor immune component (predominantly T-cells), which targets the host’s major histocompatibility complex, leading to immune-mediated toxicities across multiple systems [91]. Multiple studies have implicated gut microbial disturbances as a predictor of GVHD incidence and severity, with identification of certain differential taxa as potential predictors of GVHD [92,93,94]. For instance, a relative abundance of *Ruminococcus* and *Lactobacillus* has been associated with improved outcomes, suggesting that the dynamic microbial balance and homeostasis are again what drive immune-related toxicity.

Within the context of immunotherapy, early work demonstrated the potential impact of certain gut microbial taxa such as *Bacteroides* to mitigate irAE, such as colitis, within preclinical models in response to anti-CTLA4 therapy; when mice treated with broad spectrum antibiotics underwent reconstitution with *Bacteroides* species associated with an improved response, there was a reduction in the histopathological signs of immune colitis, pointing to an efficacy–toxicity uncoupling effect mediated by *Bacteroides* [95]. Subsequent work in a prospective group of melanoma patients treated with ipilimumab demonstrated increased representation of the *Bacteroidetes* phylum correlating with resistance to immune-mediated colitis, further supporting the hypothesis generated by the preclinical work [96]. Another prospective clinical study evaluating metastatic melanoma patients undergoing ipilimumab therapy corroborated the correlation between *Bacteroides* species and protection against immune-mediated colitis, though in this study *Bacteroides* was associated with a worse response to therapy; in contrast, enrichment for the *Firmicutis* genera was associated with an improved response and a concomitant increase in immune-mediated colitis [97].

The recent study involving a prospective profiling of the gut microbiome and dietary patterns of 103 patients in Australia and New Zealand undergoing treatment with checkpoint inhibitors demonstrated that non-responders who had severe irAEs had lower microbial diversity at baseline. Furthermore, a relative abundance of specific species, such as *Faecalibacterium prausnitzii*, was reduced in patients with irAEs (and this effect was strongest in the non-responders who developed severe irAEs), pointing again to specific perturbations of the microbiota as the potential mediator of immune related toxicity [25].

While these studies highlight the likely importance of key bacterial taxa in mediation of irAEs, the complex interplay between immune modulation and the gut microbiota have been made clear, laying the foundation for effective microbiome modulation methods to reduce irAEs in the clinical setting. Conversely, as new microbiome-based approaches are tested to augment the response to ICI, their potential concurrent impact on promoting treatment-limiting irAEs must also continue to be surveilled.

## 5. Intratumoral Microbiome—The Next Frontier in Microbiome-Based Interventions for Cancer

It is now well appreciated that microbes have a significant impact across the entire spectrum of human health and, as discussed in this review, elegant work by several groups has now highlighted the specific mechanisms through which the host microbiome drives response, resistance, and toxicity to cancer immunotherapy [2,3,5,6,81]. This newfound appreciation for the microbiome’s extensive influence in cancer progression and treatment has accordingly prompted the inclusion of polymorphic microbes as a new hallmark of cancer [98].

Although investigations into the anti- or pro-tumor effects of the intratumor microbiome are still in their infancy, recent investigations have now importantly challenged the notion of a sterile TME by demonstrating the presence of robust microbial, fungal, and viral communities within the TME [99,100,101,102,103] which may, independently or synergistically, have a profound impact on the development and strength of local anti-tumor immunity [104]. In this regard, recent advancements in spatial-omics and their use in profiling tumor specimens have unraveled new relationships between the intratumoral microbiome and local immunity. In a recent study, Nino et al. demonstrated that, in a cohort of oral squamous carcinoma and colorectal carcinoma patients, microbe-rich areas within the TME were associated with a high expression of immunosuppressive proteins such as PD-1 and CTLA-4, a greater density of monocytic/suppressor cells and fewer T-cell infiltrates, tumor cells harboring greater chromosomal abnormality, and higher cell motility as compared to regions of the TME with low microbial density [105]. Although future mechanistic efforts are needed to prove the causality of these associations, these results nonetheless suggest that the intratumor microbiome may have a dynamic effect on cancer progression, which in itself opens new avenues for translational therapies.

Thus, mining and characterizing such intratumoral microbial signatures across cancer types may have significant prognostic value in the future by helping distinguish between (i) cancerous tissues from normal adjacent tissues, (ii) different stages of a particular type of cancer, (iii) different types of cancer, and (iv) response or lack thereof to immunotherapy [99,100,101]. Although identifying microbial signatures from historic transcriptional datasets has proven challenging due to contamination and/or the selective enrichment of human transcripts from bulk tumor tissues, several bioinformatic pipelines exist and are now being developed [99,103,106] and applied to existing and/or prospectively collected cancer genomic datasets to circumvent such challenges as they pertain to profiling the intratumoral microbiome. The development of such decontamination and deconvolution algorithms is expected to foster the creation of novel interventional/translational strategies that can target or modify the intratumoral microbiome to augment the response to immunotherapy in future trials. In this regard, preliminary studies [107] and early phase human trials [108] altering the intratumoral microbiome have yielded encouraging results by demonstrating safe, feasible, and effective anti-tumor responses using such microbe-based approaches, thus setting the foundation for novel future trials aimed at improving clinical outcomes to immunotherapy.

## 6. Towards a Holistic Approach to Treat, Intercept, and Prevent Cancer

Despite significant advances, practical applications of monitoring and modulating the gut microbiome to facilitate an improved response to cancer treatment (and to improve overall health) come with challenges that need to be addressed by individuals, as well as by their treatment teams. This, however, presents an opportunity where monitoring/modulating the gut microbiome can now become a part of the holistic care provided during a lifetime—where factors such as diet, exercise, lifestyle choices, and antibiotic stewardship are given special consideration. Below, we discuss a few such factors, and anchor the impact of these factors on immunity, the microbiome, and the associated factors over the continuum of the life cycle (Figure 1).

### 6.1. Environment, Exercise, and Lifestyle

An individual’s location of residence is a critical determinant of their overall wellbeing. During the early stages of life in utero, the immunity of the mother and fetus are lowered to facilitate the tolerance of the pregnancy via expression of PD-L1 in the placenta, among other mechanisms of immune tolerance [109]. During this time, the microbiome of the fetus is shaped by the maternal microbiome, and perturbations such as exposure to broad spectrum antibiotics (and other exposures) may negatively impact the microbiome and associated immunity [110]. At the very beginning of life, the neonatal intestinal microbiome is largely influenced by the mode of delivery [111]. The microbiome, immunity, and risk of disease are further shaped by the means of feeding [112,113,114], with breastfeeding explaining the greatest amount of variance in the gut microbiome composition from months 3 to 14 of life [115]. The impact of diet and other factors on the microbiome and associated physiology continue to evolve during childhood and early adulthood, though the gut microbiome remains relatively stable once it is established, with perturbations attributed to exposure to medications (such as antibiotics) and other factors [115,116].

Especially relevant in the context of cancer, residential racial and economic segregation has a significant impact on the incidence [117,118] and mortality rates of 10 of the 12 most commonly diagnosed cancers [119]. This segregation is affected by the underlying policies that enable financial discrimination, redlining, and selective zoning to group populations based on racial and economic status [117]. These discriminatory practices further alienate the deprived population, leading to a reduced quality of life/cancer incidence of at-risk groups by way of increasing exposure to environmental carcinogens, such as atmospheric particulate matter [120], increasing financial/physiological stress and systemic inflammation [121], and reducing food security and access to transport and healthcare infrastructure [122]. Together, these external factors may independently influence the composition of the microbiome to promote disease/oncogenesis and/or prevent access to therapeutic clinical interventions (such as immunotherapy or microbiome-based interventions) following a cancer diagnosis. Re-assessment and re-evaluation of the policies and politics governing the zip code of an individual lie at the center of a possible solution for this issue and will need to be addressed by all key stakeholders in the future to reduce overall cancer incidence and mortality.

Physical inactivity and obesity are thought to be important contributors to the increasing cancer incidence worldwide, particularly among younger patients receiving a cancer diagnosis [123,124,125]. Although most of what is published in this area are cohort studies that correlate physical inactivity with cancer incidence, there are also a number of studies that correlate physical inactivity with worsened oncologic outcomes. Physical activity has been associated with a reduced risk of recurrence and improved survival in patients with breast cancer, for example [126,127,128]. A large meta-analysis reported that patients who were the most physically active had reduced cancer mortality both in the general population and among cancer survivors. This was a dose-dependent response, as they found a 13% reduction in cancer mortality for those who did moderate-intensity activity for at least 2.5 h per week and a 27% reduction for cancer survivors who completed 15 metabolic equivalents of task (MET)-hours per week of physical activity [129].

In this regard, the use of wearable technologies to monitor physical activity during/after treatment might factor as an important measure in the care provided for (pre-) cancer patients. Indeed, such wearables are increasingly being used in early clinical trials to track the data surrounding physical activity and sleep patterns and have demonstrated promising feasibility and 60–100% adherence in solid tumor patients receiving antineoplastic treatment [130]. While questions remain on how best to implement wearables into oncological practice, the rapid growth in the capabilities of such wearable technologies is encouraging and suggests that they can reliably serve as a useful adjunct for monitoring physical activity during cancer treatment in the near future.

### 6.2. Diet

Although current clinical dietary interventions have focused on high fiber [53] and fasting/ketogenic diets [59] with a demonstration of improved outcomes for patients undergoing immunotherapy and chemotherapy, respectively, there are other diet-related factors that need to be considered. For example, Western dietary patterns, red meat intake, low vitamin D intake, and excessive alcohol consumption have been linked to an increased risk of early-onset colorectal cancer [131,132,133,134]. A Western-style diet in particular, defined as a diet high in saturated fats, red meat, processed meat, sugar, and ultra-processed foods, and low in fruits, vegetables, whole grains, and fiber, has also been found to be associated with higher prostate cancer-specific and all-cause mortality [135,136]. In the same study, a prudent diet, characterized by a higher intake of vegetables, fruits, fish, legumes, and whole grains, was associated with lower all-cause mortality after a prostate cancer diagnosis [136]. Although much of oncologic care focuses on treatment planning and monitoring the disease response, oncologists now have an added responsibility to consider complementary strategies, including discussions surrounding diet and exercise, that have been shown to improve clinical outcomes and quality of life.

### 6.3. Systemic Inflammation

Dietary and other lifestyle interventions can alter the gut microbiota, and this is thought to be modulated at least in part by systemic inflammation. This is supported by the mounting evidence linking human microbiome dysbiosis to the development of an autoimmune disease in human and animal models [137,138]. Specifically, recent landmark studies have shown that high-fiber diets were associated with improved outcomes in metastatic melanoma patients treated with immune checkpoint blockade therapy, and that a dietary fiber intervention resulted in an overall decrease in the systemic inflammatory parameters with a corresponding increase in the microbiota alpha diversity [139]. Separately, another study that examined a cohort of healthy men found that dietary fiber intake was associated with increased *Clostridiales*, which have previously been shown to regulate both local and systemic inflammation. In this study, fiber intake was also associated with significantly reduced systemic C-reactive protein (CRP) in individuals without *Prevotella copri*, demonstrating novel interactions between the microbiome, diet, and inflammation [140]. These data suggest that dietary interventions and gut microbiome modulation can become key strategies in mitigating chronic inflammation, which is a key factor in tumor development and progression [141]. Future prospective data is needed to assess the markers of systemic inflammation in relation to treatment response in cancer patients.

### 6.4. Antibiotic Stewardship

Antibiotic use across the world has increased by almost 50% from 2000 to 2018 [142]. The use of antibiotics has been shown to be an independent risk factor for cancer occurrence [143] and is also associated with decreased response rates to ICI and other therapies, largely through modulation of the gut microbiome [2,3,144,145,146]. Given this data and the rise in antibiotic prescriptions globally, this is something that physicians as a whole need to pay more attention to. Although antibiotic stewardship programs (ASPs) present more challenges in the immunocompromised cancer patient, available studies indicate that the benefits of ASPs in the general populations in which they have been studied are applicable to oncology patients [147]. It is the responsibility of physicians to use sound clinical judgement in antibiotic prescribing, always considering the risks and benefits of administration with regards to the patient and oncologic outcomes.

## 7. Conclusions

Together, the gut microbiome, diet, exercise, and other factors shape overall immunity and systemic inflammation, and contribute to states of health and disease, including cancer. Though heritable genomic factors cannot be changed, there is tremendous plasticity in gut microbes and the associated immunity that can be impacted intentionally (and unintentionally). An understanding of these factors and how they impact immunity, inflammation, and overall health/risk of disease is prudent in the emerging age of precision medicine, with a holistic approach critical to cancer treatment. Certainly, with the use of novel tools, such as wearable devices and immune monitoring strategies, we will be able to realize optimal strategies to monitor and modulate gut microbes and diet to improve overall health, to combat disease, and to more effectively treat (and hopefully ultimately prevent) cancer.

## Figures and Tables

**Figure 1 cancers-15-00777-f001:**
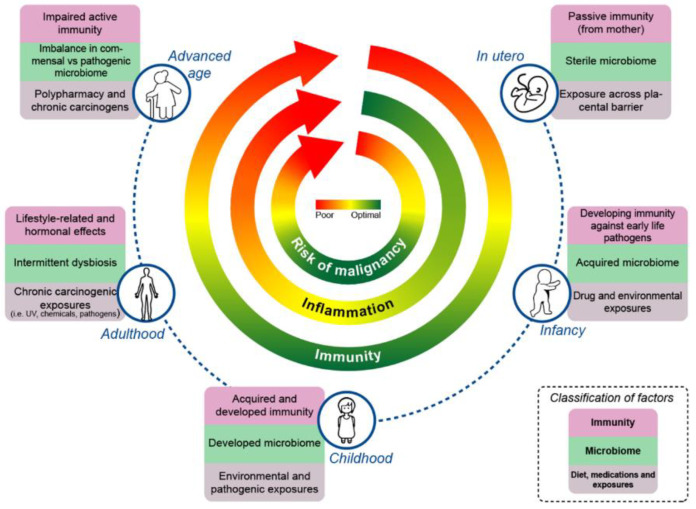
Interplay of the microbiome, inflammation, and immunity leading to a risk of malignancy across the lifespan.

**Table 1 cancers-15-00777-t001:** Trials examining the gut microbiome and the response to cancer treatment.

Disease	Stage/Disease Characteristics	Year Approved	Study Name/ NCT Number	Type of Treatment	Relevant Study Outcome (Primary or Secondary)	Status	Results
**Colorectal Cancer**							
	Metastatic or irresectable	2019	GIMICC/ NCT03941080	CT +/− TT	Response to therapy based on gut microbiome signatures	Recruiting	NA
	All stages	2021	NCT04804956	Multiple	Correlation between mesorectal microbial signatures and survival	Recruiting	NA
	Stage IV	2016	NCT02960282	5-FU based CT or anti-PD-1 IO	Correlate gut microbiome with best tumor response	Terminated (slow accrual)	NA
	Stage I-III	2022	NCT05368688	Multiple (SOC treatment)	Correlate Fusobacterium with oncologic outcomes (recurrence, metastasis, survival)	Recruiting	NA
	Unresectable, relapsed, or metastatic	2019	NCT03946917	ICI + regorafenib	Relationship between microbiome diversity and IO response	Unknown	Patients with high-abundance *Fusobacterium* have shorter PFS (2.0 vs. 5.2 months, *p* = 0.002) [14]
**Breast Cancer**							
	Newly diagnosed, HER-2 positive	2022	NCT05444647	NACT + trastuzumab	Correlation between gut microbiome and pCR	Recruiting	NA
	Newly diagnosed TNBC	2018	NCT03586297	NACT	Correlate gut and intratumoral microbiomes with pCR	Recruiting	NA
	Stage I-III	2018	NCT03702868	Adjuvant CT	Relationship between gut microbiome and DFS	Terminated (slow accrual)	NA
	Metastatic, ER+ HER2-	2020	NCT04579484	Aromatase inhibitor + CDK4/6 inhibitor	Correlate gut microbiome with time to treatment failure	Recruiting	NA
**Lung Cancer**							
	Not specified	2018	NCT03688347	CT + ICI	Correlate microbiome data (oral, nasal, skin, and gut) with ORR	Completed	Responders had increased Clostridiales (*p* = 0.018) but reduced Rikenellaceae (*p* = 0.016) in gut microbiome [35]
	Stage IIIB/IV NSCLC with 1 or 2 prior systemic therapies	2017	NCT03195491	Nivolumab (2nd/3rd line)	Correlate gut microbiome signatures with clinical outcomes	Completed	- High alpha diversity (Shannon index > 2.31)- improved PFS (HR 4.2 on multivariate analysis) - *Alistipes putredinis, Prevotella copri,* and *Bifidobacterium longum* enriched in responders [9]
	Stage III NSCLC	2021	NCT04711330	Chemo/RT followed by maintenance durvalumab	Correlate microbiome (throat and stool) with cancer progression during IO treatment	Recruiting	NA
	Stage IIIB-IV NSCLC	2021	NCT04954885	Pembrolizumab +/− CT	Correlate gut microbiome with OS and PFS	Recruiting	NA
	Inoperable stage III NSCLC	2021	PRECISION/NCT05027165	Chemo/RT followed by durvalumab	Correlate gut/saliva microbiome with 12 and 24mo PFS	Recruiting	NA
	Stage II-III NSCLC, newly diagnosed	2019	NCT04013542	RT + combination ICI	Correlate microbiome to clinical outcomes (ORR, PFS, OS)	Recruiting	NA
	Stage IV NSCLC	2021	NCT04909034	Pembrolizumab + MS-20	Correlate gut microbiome and clinical outcomes	Recruiting	NA
	Stage IV or recurrent NSCLC	2020	NCT04636775	ICI	Microbiome differences between responders vs. non-responders	Recruiting	NA
	Metastatic NSCLC, failed at least 1 prior treatment	2017	NCT03168464	RT + combination ICI	Correlate microbiome changes with ORR	Completed	NA
**Pancreatic Cancer**							
	Pancreatic ductal adenocarcinoma	2021	PDA-MAPS/NCT04922515	Not specified	Associate intestinal and tumoral microbiome with treatment response	Recruiting	NA
**Gynecologic Cancer**							
	Advanced or recurrent	2021	NCT04957511	IO	Examine whether the gut microbiome is associated with the response to cancer immunotherapy	Recruiting	NA
**Melanoma**							
	Stage IV	2021	NCT05102773	ICI	Correlate microbiome alpha-diversity with response to treatment	Active, not recruiting	*Ruminococaceae* associated with development of a potential irAE (*p* = 0.03) [36]
	Stage III-IV	2018	PRIMM/NCT03643289	IO	Correlate gut microbiome diversity with response to treatment	Recruiting	- *Bifidobacterium pseudocatenulatum, Roseburia* spp. and *Akkermansia muciniphila* associated with responders - Limited reproducibility of microbiome-based signatures across cohorts with machine learning [37]
	Stage III cutaneous melanoma	2016	OpACIN-neo/NCT02977052	Neoadjuvant ipilimumab + nivolumab	Associations between gut microbiome with response rates and toxicity	Active, not recruiting	- *Faecalibactrium prausnitzii, Butyricicoccus pullicaecorum,* and *Akkermansia muciniphilia* significantly enriched in responders - Reduced *F. prausnitzii* associated with severe irAEs [25]
**Glioblastoma Multiforme**							
	Not specified	2018	NCT03631823	RT +/− CT (with temozolomide)	Correlate gut microbiome and PFS	Unknown	NA
	WHO grade 4, newly diagnosed	2022	THERABIOME-GBM/NCT05326334	CT + RT	Gut microbial composition in late versus early progressors	Not yet recruiting	NA
**Head and Neck Cancer**							
	Unresectable locoregionally advanced disease	2021	COMRAD-HNSCC/NCT05156177	Definitive RT	Compare fecal microbiome between responders and non-responders	Recruiting	NA
**Esophageal Cancer**							
	Stage I-III SCC	2022	NCT05199649	NACT + Sintilimab	Correlate gut microbiome and metabolic markers with treatment efficacy	Recruiting	NA
**Multiple Cancer Types**							
	Hematologic and solid malignancies	2021	NCT05112614	Multiple, including SCT	Correlate gut microbiome with clinical response	Recruiting	NA
	Stage III/IV NSCLC, colorectal, TNBC, pancreas	2020	ARGONAUT/NCT04638751	CT and/or IO	Correlate gut microbiome with treatment response	Recruiting	NA
	Advanced melanoma, RCC, and NSCLC	2019	MITRE/NCT04107168	IO	Correlate microbiome signature with PFS of 1 year or greater	Recruiting	NA
	Not specified	2020	ONCOBIOTICS/NCT04567446	Multiple (CT, HT, IO)	Define metagenomic signatures associated with effectiveness of anticancer therapies (ORR, PFS, OS)	Recruiting	In NSCLC cohort treated with ICI, relative abundance of *Akkermansia* associated with 10% improvement in ORR. [6]
	Melanoma, NSCLC, RCC, TNBC	2021	NCT05037825	ICI	Association between the gut microbiota and ICI treatment efficacy	Recruiting	NA
	Advanced solid tumors	2019	INSPECT-IO/ NCT04107311	IO combination	Correlate gut microbiome with toxicity and cancer outcomes	Recruiting	NA
	Advanced solid tumors	2019	NCT04114136	ICI +/− metformin or rosiglitazone	Differences in composition of oral and stool microbiomes between responders and non-responders	Recruiting	NA
	Advanced solid tumors	2019	NCT04204434	ICI	Correlate gut microbiome with response to treatment	Recruiting	NA

CT = chemotherapy. NACT = neoadjuvant chemotherapy. TT = targeted therapy. IO = immunotherapy. ICI = immune checkpoint inhibitor. HT = hormone therapy. SCT = stem cell transplant. NSCLC = non-small cell lung cancer. TNBC = triple negative breast cancer. RCC = renal cell carcinoma. pCR = pathologic complete response. DFS = disease-free survival. PFS = progression-free survival. OS = overall survival. ORR = overall response rate. SOC = standard of care. irAEs = immune-related adverse events. NA = results not published.

**Table 2 cancers-15-00777-t002:** Current trials in diet/supplements and probiotics/prebiotics in cancer and cancer therapy response.

Class of Intervention	Specific Intervention	Type of Therapy	Tumor Type Treated	NCT	Route	Preliminary or Final Results
High fiber	High-Fiber Diet	Pembrolizumab/Nivolumab	Melanoma	NCT04645680	NA	NA
	High fiber, exercise	Anti-PD1	Melanoma	NCT04866810	NA	NA
	High fiber, exercise	ICB	Multiple types	NCT04866810	NA	NA
	High fiber, exercise	ICB	Multiple types	NCT04866810	NA	NA
	High fiber, exercise	ICB	Multiple types	NCT04866810	NA	NA
	Leafy greens and vegetables	NA	Prostate cancer	NCT01238172	Oral	Behavioral intervention encouraging leafy green and vegetable consumption did not reduce the risk of prostate cancer progression in men [71]
Diet intervention	Diet intervention	Chemotherapy	Breast cancer	NCT03314688	NA	NA
	Diet intervention	Endocrine therapy	Breast Cancer	NCT04079270	NA	NA
	NutriCare Plus: Meal intervention	Surgery and/or systemic therapy and/or radiation	Lung cancer	NCT04986670	NA	NA
Ketogenic diet	Ketogenic diet	NA	Mantle cell lymphoma	NCT04231734	NA	NA
	Ketogenic Diet	Ipilimumab/Nivolumab	Renal cell carcinoma	NCT05119010	NA	NA
Fasting	Prolonged Nightly Fasting	SOC ICB	Head and neck squamous cell carcinoma	NCT05083416	NA	NA
	Intermittent Fasting	NA	Chronic lymphocytic leukemia/small lymphocytic lymphoma	NCT04626843	NA	NA
	Prolonged Nightly Fasting	ICB	Head and neck squamous cell carcinoma	NCT05083416	NA	NA
Other supplements	Vitamin D	Neoadjuvant chemotherapy	Breast cancer	NCT04677816	Oral	NA
	Grape Seed Extract	NA	Prostate cancer	NCT03087903	Oral	Preliminary results indicate that 300 mg of daily grape seed extract may improve PSA kinetics in patients with a rising PSA after maximum local therapy [72]
	Fermented Soybean Extract	Pembrolizumab	Non-small cell lung cancer	NCT04909034	Oral	NA
	Resistant starch foods	NA	Colorectal cancer	NCT03781778	Oral	NA
	Fish oil	NA	Colorectal cancer	NCT01661764	Oral	No difference in proliferative or apoptotic markers in rectal mucosa [73]
Prebiotic	Prebiotic	NA	HSCT	NCT04629430	Oral	NA
	Prebiotic + Probiotic	Chemotherapy + radiation	Anal squamous cell cancer	NCT03870607	Oral	NA
Probiotics	Probiotics	NA	Breast and lung cancer	NCT04857697	Oral	NA
	Probiotics	NA	Colorectal cancer	NCT03782428	Oral	Significant reduction in proinflammatory cytokines among CRC patients taking probiotics vs. placebo [74]
	Probiotics	PD-1 inhibitor	Liver Cancer	NCT05032014	Oral	NA
	Probiotics	PD-1 inhibitor + chemotherapy	Non-small cell lung cancer	NCT04699721	Oral	NA
	Probiotics	PD-1 inhibitor	Renal cell carcinoma	NCT03829111	Oral	NA
	Probiotics	EGFR inhibitor	Non-small cell lung cancer	NCT01465802	Oral	Probiotic did not have any impact on adverse event profile after treatment with dacomitinib [75]
Specific strains	Lactobacillus Bifidobacterium V9 (Kex02)	Carlizumab with platinum	Non-small cell lung cancer	NCT05094167	Oral	NA
	Lactobacillus rhamnosus GR-1, Lactobacillus reuteri RC-14	NA	Breast cancer	NCT03290651	Oral	NA

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
