# Peer review of "Monitoring and Modulating Diet and Gut Microbes to Enhance Response and Reduce Toxicity to Cancer Treatment"

_cancers, 2023, doi:10.3390/cancers15030777_

Round 1

Reviewer 1 Report

 Knisely et al studied and wrote the review of the role of gut microbe in cancer treatment and management. It is increasingly being advanced that gut microbiome plays important role in cancer. The review is written nicely with information that would be useful to the reader. This is a vast area and difficult to cover every aspect.

Main comments

I suggest authors should consider a method section describing PRISMA guidelines to present a review. It will give more authenticity and acceptable format for accepting information described in the manuscript. It will also improve the manuscript considerably.

Reviewer 2 Report

In this manuscript of "Monitoring and modulating diet and gut microbes to enhance

response and reduce toxicity to cancer treatment", authors figured out that an association between particular gut microbiome signatures and improved response to immune checkpoint blockade, chemotherapy, and adoptive cell therapies, as well as the role of particular microbes in driving treatment-related toxicity and how the microbiome can be modulated through strategies such as fecal transplant. In addition, the MS summarizes that high fiber and ketogenic diets in improved response rates to immunotherapy and chemotherapy, respectively. It enlightens that the potential of a holistic approach to cancer treatment might have reference significance. A lot of work was done and it is certainly contributing to the current scientific knowledgebase. But this needs revision and the author reply comments properly before accepted. The comments and questions are as follows:

1. When “ORR” first shows in the manuscript it should not be abbreviation.

2. Is there any detail about promoting ICI response through the secretion of metabolites that serve key functions in immune system modulation reported? The mechanism of how gut microbes affecting immune system and ICIs should be added.

3. The example in the section of Targeted and other therapies could not clarify how the gut microbiome plays the role in response to cancer treatment well.

4. There is a repeated title Immunotherapy in page 10. Please check all the titles in this MS carefully and the format need reconfirmation and revision.

5. It is noted that this manuscript needs careful editing and paying particular attention to the review structure, also, improving headings to make each part clearly and logical.

6. A holistic approach to cancer treatment authors advocated should be provided in Conclusion.

Reviewer 3 Report

This review is very detailed and rigorous with scientifically sound knowledge about the association of microbiomes with various kinds of cancer treatments and their outcomes. They have done a good job of describing the effects of microbiomes (including various taxa), different kinds of diet, exercise, antibiotics, etc. concerning various cancer treatments. They have highlighted all relevant studies/literature that implicate microbiomes in cancer progression and treatments. The authors also shed light on a very important aspect of tumorigenesis i.e. intratumoral microbiome, which might prove an important hallmark for cancers, and could open new therapeutic options. This review will be very useful for people in the cancer field. However, certain issues need to be addressed to improve the quality of this review further.

1. This review seems to be written for people in the cancer field only. There are a lot of jargons and specific terms used in this paper, which makes it difficult for people outside of the field to understand it.

2.  The full form of abbreviated terms was missing. e.g. IL-6, IFN-1, TNFα, HER2, CTLA4, TME etc. should be mentioned wherever applicable. The full form of ORR was mentioned on page 8 (table 1), whereas it was first mentioned on page 3. PD-1 has been mentioned at various places throughout the paper, but the authors forgot to introduce it or at least mention its full form. I know it's a common programmed cell death protein, but it is worth putting its full form for beginners in the field.

3.  Authors need to pay more attention to the titles and subtitles. Titles and subtitles need to be more structured to convey the message clearly to the readers. The first title is in bold whereas the rest of the subtitles (or titles) are in regular font. Does that mean that there is only one title, and the others are all subtitles? The first title” Role of the gut microbiome in response to cancer treatment” and subtitles like “immunotherapy” “chemotherapy” and “targeted and other therapies” seem fine as a part of one section. However, are the next sections like “impact of antibiotics” and “modulating gut microbes to improve outcomes” continuation of the first title or they are independent? They should be put in bold font if they are independent titles.

4.  “The role of the diet in response to cancer treatment “should be an independent title and could have better subtitles like “effects of high fiber diets” “ effects of fasting” “effects of ketogenic diets” etc. instead of redundant subtitles like “immunotherapy” “ chemotherapy and other therapies”.

5.  On page 9, the authors mentioned” modulation of the gut microbiome can be accomplished in a variety of ways, including the introduction of antibiotics, probiotics and prebiotics, dietary interventions, and fecal microbiome transplantation (FMT)”.  I expected them to shed light on individual ways mentioned above, however, the rest of the three paragraphs are solely focused on FMT. They could have subsections like “effects of antibiotics” “prebiotics and probiotics” dietary inventions” and “FMT” as subsections and describe their effects on cancer treatments. The information given in the “impact of antibiotics” can be moved to the “effects of antibiotics” subsection. In my opinion, having more structured subtitles will help readers to grasp the knowledge in a much clearer and better way.

6.  Table 1 has a “status” column where lots of statuses are “Recruiting”. What does it mean? Does recruiting mean the study is still ongoing or in progress? Most of the results column has “NA”, which doesn’t add anything to the information given and occupies more space. Is it possible to incorporate the information given for those seven studies to include with the “Relevant study outcomes” column?

7.  What does * mean in table 2? There is no mention of the meaning of * either in the table or figure legends. Does that mean that the references are not available?

8. There is no numbering of lines till page 16. Numbering should start from the beginning of the paper.

9. Use of complex English words like typify, efficacious, refractory, orointestinal, post-hoc, ameliorate, stewardship, polypharmacy, adjunct, etc. should be avoided and could be replaced with simpler words to allow the ease of reading. The authors should keep laymen in their mind while writing the review, so it will be beneficial to a larger group of researchers as well as to the general public.

10.  Better proofreading is recommended to avoid some minor issues like spelling mistakes “profling” in line 118, spacing issues like lack of spacing in lines 212, 213 and more spacing in line 225, Page 4: last line of first paragraph mentioned “…response to standard of care ovarian cancer therapy”. This sentence needs to be rewritten to convey the message clearly.

Round 2

Reviewer 1 Report

The manuscript may be accepted after English editing and spell check.